# Falls in young adults: The effect of sex, physical activity, and prescription medications

HyeYoung Cho[1,2], Michel J. H. Heijnen[3], Bruce A. Craig[4], Shirley Rietdyk[1,2]*

1 Department of Health and Kinesiology, Purdue University, West Lafayette, IN, United States of America,
2 Center for Aging and the Life Course, Purdue University, West Lafayette, IN, United States of America,
3 School of Health and Applied Human Sciences, University of North Carolina Wilmington, Wilmington, NC, United States of America, 4 Department of Statistics, Purdue University, West Lafayette, IN, United States of America

* srietdyk@purdue.edu

**Data Availability Statement:** The data underlying the results presented in the study are available from https://www.kaggle.com/shirleyrietdyk/falls-in-young-adults-cho-et-al-2021-plos-one.

## Abstract

Falls are a major public health issue not only for older adults but also young adults, with fall-related injuries occurring more frequently in adult females than males. However, the sex differences in the frequency and circumstances of falls in young adults are understudied. This research quantified the frequency and circumstances of falls as a function of sex, physical activity, and prescription medications in young adults. For 16 weeks, young adult participants (N = 325; 89 males; 19.9±1.1 years) responded to a daily email asking if they had slipped, tripped, or fallen in the past 24 hours. Falls and fall-related injuries were not uncommon in young adults: 48% fell at least once, 25% fell more than once, and 10% reported an injury. The most common activities at the time of the fall for females were walking (44%) and sports (33%), and for males, sports (49%) and walking (37%). A zero-inflated Poisson model revealed that higher number of falls were associated with the following: higher levels of physical activity ($p = 0.025$), higher numbers of medications ($p<0.0001$), and being male ($p = 0.008$). Regarding circumstances of falling, females were more likely to be talking to a friend at the time of the fall (OR (95% CI): 0.35 (0.14–0.73); $p = 0.01$). For slips and trips without a fall, males and females reported the same number of slips (OR (95% CI): 0.885 (0.638–1.227) $p = 0.46$), but females reported more trips (OR (95% CI): 0.45 (0.30–0.67); $p<0.01$). Only females reported serious injuries such as concussion and fracture. In conclusion, the rate of falls in young adults was affected by physical activity levels, number of medications, and sex. Quantifying and understanding these differences leads to increased knowledge of falls across the lifespan and is instrumental in developing interventions to prevent falls.

## Introduction

Recent research has demonstrated that falls are a serious concern not only for older adults, but also for young adults [1–6]. The total cost of fall-related injuries for adults aged 18–24 in the US exceeded $7 billion in 2010 [6]. The majority of research in young adults has examined

**Funding:** Publication of this article was funded in part by Purdue University Libraries Open Access Publishing Fund. The funders had no role in study design, data collection and analysis, decision to publish, or preparation of the manuscript. There was no additional external funding received for this study.

**Competing interests:** The authors have declared that no competing interests exist.

fall-related injuries obtained from medical reports [1, 3] and injury self-report [2]. However, these approaches do not capture all falls, only those with injuries. A more complete understanding of falls in young adults can be obtained by quantifying all falls, including non-injurious and injurious falls.

While it is well-known that older females sustain more injuries than older males (Fig 1; [7, 8]), the higher injury rate for females is first evident at age 20 (Fig 1 inset). In young adults aged 20–45 years, females were more likely to report a fall than males (20 vs 17%, respectively), and were more likely to be injured from the fall (81 vs 61%, respectively) [9]. Extending these analyses to a more restricted age range will provide insights at the age when differences in fall injuries are first noted between females and males (20 years, Fig 1 inset). Examining sex-related differences in falls in young adults is especially important as young females (aged 20–29 years) had a 25% increase in fall-related fractures in the 10 year period from 2000 to 2010, while in males of the same age, the increase was 5% (not statistically significant) [1]. Furthermore, females had a higher fall-related injury rate on stairs at all ages except ≤ ten years [3], and the proportion of fall-related injuries on stairs was highest among young females [2].

Young adults are more active than older adults [10], and a higher activity level in young adults was associated with a higher frequency of falls [4]. This likely results from higher sports participation in young adults [11], which increases exposure to hazardous situations [12]; this increased risk is supported by the observation that sports-related falls are common in young adult males [2, 9]. Conversely, for older adults, higher levels of physical activity are generally considered as protective against falls (e.g. [13]). Quantifying the association between physical activity and falls in young adults will increase our understanding of the factors related to falls across the adult lifespan.

Another factor that may be relevant to falls in young adults is number of prescription medications. Prescription medications have been linked to falls in older adults (e.g. [14, 15]), and to fall-related injuries in adults aged 25–60 years [16]. While the latter study includes young adults, the age range spans four decades. It is important to examine a more restricted age range, to determine if the association between number of medications and falls is also evident when only young adults are included in the sample.

Overall, while previous research has demonstrated that sex, physical activity, and number of medications are associated with falls and fall-related injuries in young adults [2–6, 9, 16], no

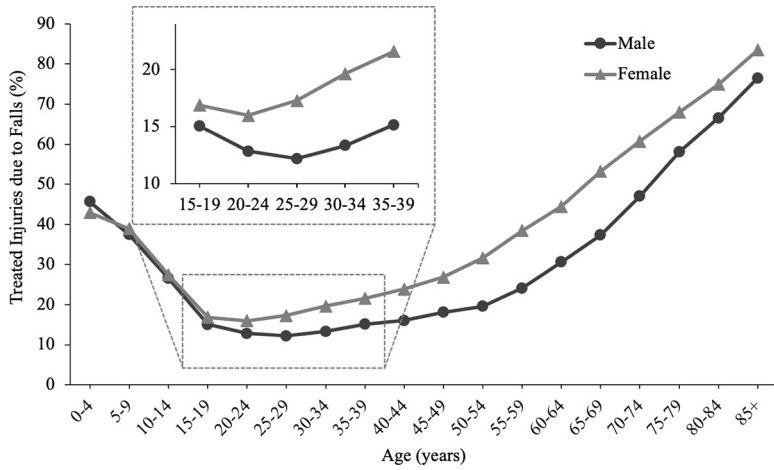

**Fig 1. Percent of all medically treated injuries that resulted from falls as a function of age and sex for 2015.** Data extracted from the CDC's Web-based Injury Statistics Query and Reporting System (WISQARS[TM]) for non-fatal injuries [7]. Inset shows greater detail for ages 15–39 years.

studies have examined these factors concurrently. The primary purpose of this study was to determine if the frequency of falls was associated with sex, physical activity level, and number of medications in college-aged young adults. Fall frequency was quantified with a daily on-line survey [4] as contemporaneous fall reports given at regular intervals (i.e. prospective) are more accurate than retrospective reports [17–20]. The daily survey also allowed us to quantify fall circumstances (e.g. activity at time of falls, injury), and slips and trips that did not result in falls. While slips and trips are common precipitating causes of falls and fall-related injuries across the adult lifespan [2, 4, 19, 21–23], the frequency of slips and trips during daily activities is understudied. These observations in young adults will increase knowledge that may lead to new intervention targets to mitigate the high costs associated with falls.

## Methods

Undergraduate students (N = 343; 96 males) participated in this study for one academic semester (16 weeks). The undergraduate students enrolled in the department (Health and Kinesiology) are 64% female, consistent with the higher percentage of female participants (72%) observed in the study. The results from the first 94 subjects were previously published [4]. All students were taking the same course, and were offered 1% extra credit for (A) participating in the survey or (B) completing a short written assignment. The goal of the alternate assignment was to limit potential coercion to participate in the research. Of the 510 students enrolled in the course over nine semesters (from 2014–2019), 343 chose to participate in the survey (67%) (Fig 2). No instruction regarding required participation in the daily surveys was provided; all students who signed the online consent received 1% extra credit whether or not they responded to the daily emails. The response rate of the 343 participants ranged from 0 to 100%, with mean, median, and mode of 91%, 98%, and 100%, respectively (response rate was calculated for each participant as (number of responses/number of daily surveys*100)). Weekly response rate is shown (Fig 3). Eighteen participants (7 males) were excluded from further analyses because they responded to less than 35% of the daily emails [4] (Fig 2); six of these 18 participants did not respond to any daily surveys after signing the online consent form. In the remaining 325 participants, the response rate ranged from 38–100%. These 325 participants (age 19.9±1.1 years, range 18–27 years, 89 males; height 169.9±9.3 cm; weight 68.9±15.1 kg) were included in the following results. The informed consent and study methods were approved by the Human Research Protection Program.

Two types of surveys were distributed via Qualtrics Survey Software (Qualtrics Labs Inc. Provo, UT): (1) initial survey distributed once for consent and demographic information and (2) daily survey distributed daily for 16 weeks. These surveys are available as supplemental content in [4]:

### Initial survey (distributed once in week 1)

The initial survey included demographics (e.g. age, sex, height, weight, number of prescription medications, physical activity level), and was distributed by email once in the first week. For the number of prescription medications, participants were instructed to select one of the follow six categories: 0, 1, 2, 3, 4, 5 or more. Two subjects (females) selected 5 or more. Since we could not determine the exact number of medications, these two subjects were removed from analyses involving medication. The number of prescription medications, rather than drug name or category, was selected for several reasons. First, many patients cannot name any of their medications [24], and if students completed the on-line survey away from home, medication information may not have been available. Second, the number of prescription medications were associated with increased risk of fall-injury in young and middle-aged adults [16]. Third,

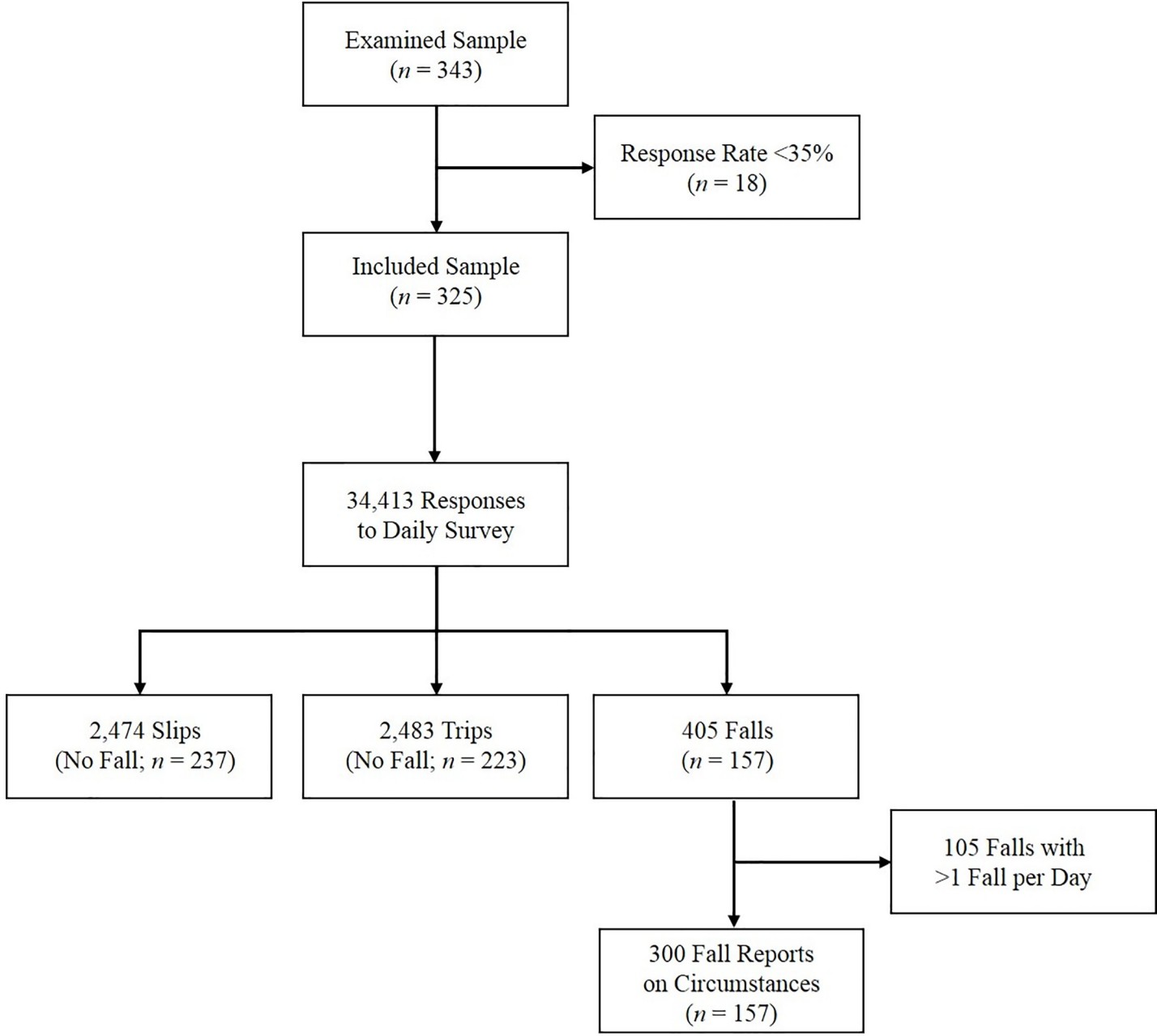

**Fig 2. Participant flow diagram.** Figure also includes the number of responses to the daily survey, number of slips and trips reported without a fall, number of falls, and number of reports on fall circumstances.

the number of prescription medications increased fall-risk even after adjusting for fall-risk-inducing drugs in older adults [15]. Therefore, in this study, the number of prescription medications were quantified, not medication name or category. Physical activity was assessed with the Leisure-Time Exercise Questionnaire (LTEQ) [25]. The LTEQ is a short survey that quantifies frequency of participation in three categories of physical activity (strenuous, moderate, and mild) with at least 15 minutes duration for an average week. The LTEQ provides a single score which ranks physical activity levels in individuals [26].

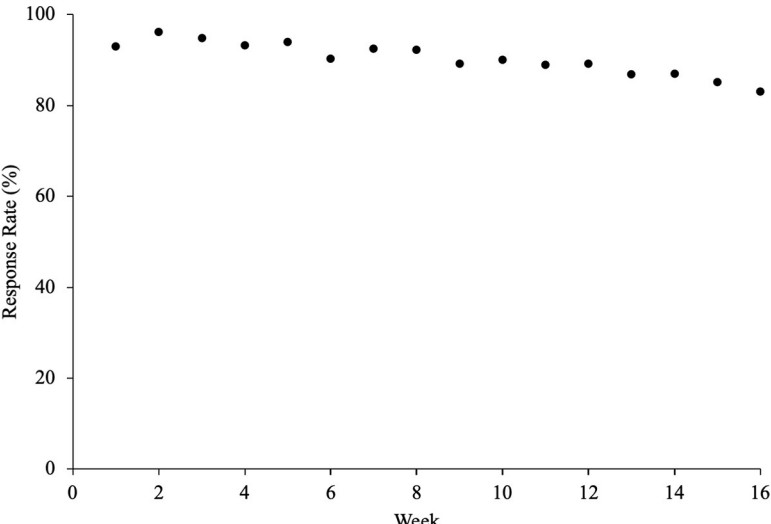

**Fig 3. Survey response rate as a function of week.** Eighteen participants who had a response rate less than 35% are included.

## Daily survey (distributed daily for 16 weeks)

Each morning, participants received an emailed survey that asked "Have you slipped, tripped or fallen in the past 24 hours?". The following definitions were provided with the question: "FALL–an undesired contact of any body part (other than the feet) with the ground or a lower surface" [27]; "TRIP–your foot or shin contacted the ground or other object unexpectedly"; and "SLIP–your foot slipped while in contact with the ground". If the participant responded 'No', the survey concluded. If 'Yes' was selected, the participant was asked to enter the number of occurrences of each trip, slip, or fall in the past 24 hours. When a fall was reported, questions regarding the circumstances were also presented. Circumstances included activity, precipitating cause, time of day, injuries and other factors. If the participant reported more than one fall in the previous 24 hours, participants were asked to describe the circumstances of the fall that were perceived as the most serious or interrupted their movement the most. Limiting the survey to one fall a day was done to mitigate survey fatigue observed in pilot research.

## Data analyses

The frequency of falls was quantified with two approaches: (1) with all falls included and (2) with sports-related falls excluded. The latter categorization (sports-related-falls-exclusion analysis) was completed because sports are associated with a higher risk of falls, and avoiding falls during sports may be impossible. For the sports-related-falls-exclusion categorization, when more than one fall occurred in a day, it was assumed that all falls on that day occurred for the same reason as the one fall that had an associated fall report. Thus, if two falls occurred on one day, and the reported fall was due to sports, both falls were eliminated in the sports-related-falls-exclusion categorization.

## Statistical analyses

**Fall frequency.** We used a zero-inflated Poisson model to describe the relationship between number of falls and sex, activity level (LTEQ score), and number of prescription medications. We selected a zero-inflated Poisson model because of a larger than expected number

of participants (under the Poisson distribution) with zero falls. The model included two and three-factor interactions (sex, physical activity, and number of medications) and we used AIC to select the best model. The LTEQ and number of medications were centered (the mean of all values was removed from each individual value). Centering was completed in order to reduce the collinearity between the interaction term and the associated main effect terms. The number of response days for each participant (up to 112 responses: 7 surveys a week for 16 weeks) was used as an offset variable to account for differences in sampling intensity.

**Fall circumstances.** Three hundred fall reports were obtained from 157 participants (Fig 2), indicating that some participants contributed more than one fall report. Fall circumstances (e.g. activity at time of fall, cause of fall, injury) were quantified for all participants and also separately for males and females. A bias-corrected and accelerated (BCa) bootstrap interval of the odds ratio (OR; males divided by females) was constructed (bivariate analysis). The BCa analysis accounted for the fact that each participant could contribute multiple falls (Fig 2). Confidence intervals were not reported for cases when the count for either sex was less than 5, or when the category was "other."

**Slip and trips without falls.** The probability of reporting a slip or a trip *without an associated fall* was examined with a mixed logistic regression, and the OR was determined for males versus females (males divided by females).

The zero-inflated Poisson model and mixed logistic regression were completed in SAS 9.3 (Cary, NC, USA), and the BCa bootstrap interval of the OR was completed in R (RStudio, Boston, MA, USA). Significance level was set at $p \leq 0.05$.

## Results

### Response rate of 343 participants

The response rate decreased from 95% (first two weeks) to 84% in the last two weeks, with an average of 90% (Fig 3). Male response rate was not different from females (89 and 91%, respectively; OR (95% CI): 1.25 (0.70–2.25); p = 0.45). The following results do not include the 18 subjects with response rates lower than 35%, as described in the methods. From the 325 subjects, a total of 34,413 responses to Question 1 of the daily survey were obtained (Did you slip, trip or fall in the past 24 hours?) (Fig 2).

### Fall frequency: Descriptive summary

Four hundred and five falls were reported by 157 subjects (48% of 325 subjects) (Fig 2), with 115 falls reported by 43 males (48%) and 290 falls reported by 114 females (48%). A completed fall-circumstance report was available for 300 of the 405 falls (74% of all falls); a fall-circumstance report was not completed for 105 falls (26%) due to participants falling more than once in a day (see methods). There were 34 reports of two falls in one day, and 21 reports of more than two falls in one day. Eighty-one subjects (25%) fell more than once in the 16-week interval (termed frequent fallers). When examined as a function of sex, 43 males (48%) and 114 females (48%) fell at least once, and 24 males (27%) and 57 females (24%) fell more than once (Fig 4A).

### Fall frequency: Effect of sex, physical activity, and number of medications

For *frequency of falls*, we started with a zero-inflated Poisson model that included two and three-factor interactions and used AIC to select the best model. This resulted in all three variables in the Poisson component: sex (p = 0.008), physical activity (LTEQ) (p = 0.025), and number of medications (p<0.0001), and only the number of medications in the zero-inflation

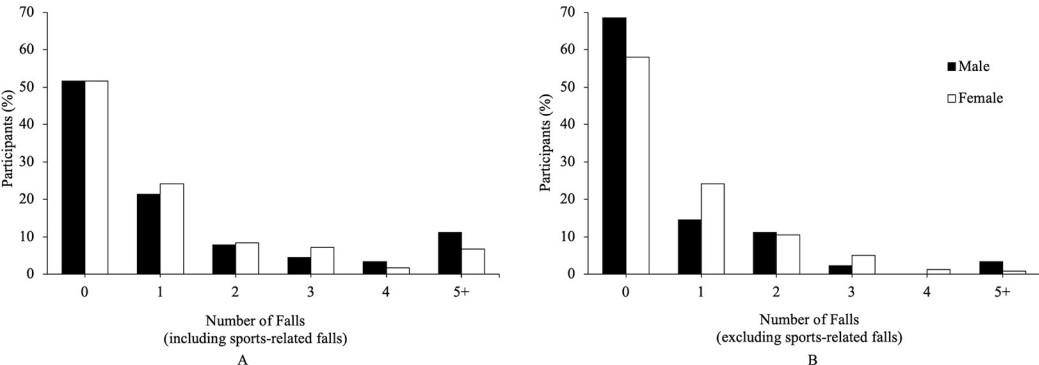

**Fig 4. Percentage of participants versus the number of reported falls with sports-related falls included (A) and sports-related falls excluded (B).** Eighteen participants with a response rate less than 35% are not included.

component ($p$ = 0.025) (See Table 1 for parameter estimates and CI). Holding the other two factors fixed, a male's average number of falls was 38% higher (95% CI: 8.0–89.0%) relative to a female, and an increase of 10 points on LTEQ increased the mean number of falls by 4% (95% CI: 0.7–8.0%) (Ten LTEQ points would be approximately equivalent to 15 minutes each week of strenuous exercise, 30 minutes of moderate exercise, or 45 minutes of light exercise). As noted above, number of medications was significant in both Poisson ($p<0.0001$) and zero-inflated distributions ($p$ = 0.025) (Fig 5), indicating that while the mean number of falls increased with number of medications, this effect was dampened by a greater chance of zero falls. Thus, the mean number of falls increased 23% for 0 to 1 medications, 21% (1 to 2 medications), 19% (2 to 3 medications), and 16% (3 to 4 medications).

### Falls that were not associated with sports: Descriptive summary

Since fall-risk is higher during sports activities, fall frequency was also examined without sports. Two hundred and twenty-two non-sports-related falls were reported by 127 subjects (39% of 325 subjects), with 57 subjects (18%) reporting more than one fall (Fig 4B). When examined as a function of sex, 28 males (31%) and 99 females (42%) fell at least once, and 15 males (17%) and 42 females (18%) fell more than once (Fig 4B).

To determine if the frequent fallers were more likely to fall during sports, the activity at the time of the fall was examined as follows. Sixty-seven of the 81 frequent fallers (83%) reported at least one fall that was not related to sports. Thirty-nine of the 81 frequent fallers (48%) had no falls related to sports. Fifteen of the 81 frequent fallers (19%) reported 100% of their falls as due to sports.

### Causes of falls

The four main causes of falls were slip (38%), trip (29%), hit/bump (11%), and loss of support with object (7%) (Table 2). When an environmental cause was selected, ice/slippery surface

**Table 1. Parameter estimates and 95% CI for zero-inflated and Poisson components of the zero-inflated Poisson model.**

|  | Zero-Inflation Component | | | Poisson Component | | |
|---|---|---|---|---|---|---|
|  | **Estimate** | **95% CI** | **$p$** | **Estimate** | **95% CI** | **$p$** |
| Intercept | -0.3066 | -0.6564–0.0432 | 0.0858 | -3.9832 | -4.1351– -3.8313 | **<0.0001** |
| Medications | 0.3214 | 0.0413–0.6015 | **0.0245** | 0.3367 | 0.2349–0.4384 | **<0.0001** |
| Sex |  |  |  | 0.3529 | 0.0926–0.6132 | **0.0079** |
| Physical Activity |  |  |  | 0.0041 | 0.0005–0.0077 | **0.0250** |

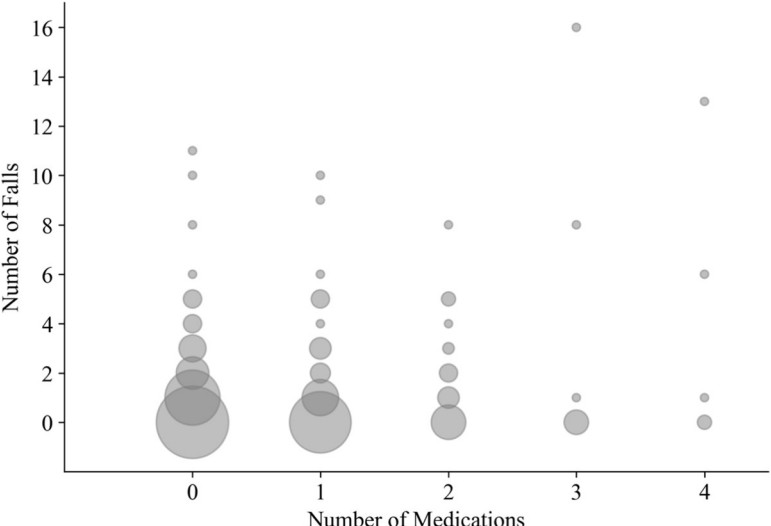

**Fig 5. Bubble plot of the number of medications versus the number of falls.** With a zero-inflated Poisson model, number of medications was significant in both Poisson ($p < 0.0001$) and zero-inflation distributions ($p = 0.025$) (see also Table 1). While the mean number of falls increased with number of medications, this effect was dampened by a greater chance of zero falls: mean number of falls increased 23% for 0 to 1 medications, 21% (1 to 2 medications), 19% (2 to 3 medications), and 16% (3 to 4 medications).

(34%) and stairs (13%) were the main environmental causes for all falls. When only non-sports related falls were considered, ice/slippery surface (44%) and stairs (22%) were the main environmental causes. The majority of ice/slippery surface falls occurred outdoors (88%), while the majority of stair-related falls occurred indoors (69%). Stair-related falls occurred at a similar percentage for ascent (52%) and descent (48%).

### Activity at time of fall

The top three activities for males were: sports, walking, running/jogging; for females: walking, sports, running/jogging (Table 2). Thirty-five percent of falls occurred while multitasking. The most common multitask was talking to a friend, which occurred in 25% of the falls. Few falls occurred while texting (2%) or talking on phone (1%) (Table 2). Of the six falls during texting (five females, one male), three occurred on stairs and two on icy/slippery surfaces.

### Injuries from falls

Forty-three falls in 34 participants resulted in an injury (14% of falls, 10% of participants) (Table 2). Three falls resulted in a serious injury (broken bones or concussion) (1% of falls).

### Falls related to alcohol and drugs

Alcohol or drugs were reported to be related to the fall in 33 falls (11% of falls).

### Other circumstances associated with falls

The majority of falls occurred outdoors (57%) and in bright light circumstances (66%). The most common (41%) time when falls occurred was between 6 pm and 12 am.

**Table 2. Fall circumstances for all fallers, and comparison for male versus female fallers.**

| | Number of Falls (percent) | | | OR | |
|---|---|---|---|---|---|
| | Total Fallers (N = 157, 300 falls) | Male Fallers (N = 43, 73 falls) | Female Fallers (N = 114, 227 falls) | | 95% CI |
| **Cause of falls:** | | | | | |
| Slip | 113 (38) | 29 (40) | 84 (37) | 1.122 | 0.576–2.266 |
| Trip/stumble | 87 (29) | 23 (32) | 64 (28) | 1.172 | 0.503–2.406 |
| Hit/Bump | 33 (11) | 11 (15) | 22 (10) | 1.653 | 0.453–4.743 |
| Loss of Support with object | 20 (7) | 5 (7) | 15 (7) | 1.039 | |
| Other | 47 (16) | 5 (7) | 42 (19) | | |
| **Injuries:** | | | | | |
| Injured | 43 (14) | 7 (10) | 36 (16) | 0.563 | 0.396–1.318 |
| Contusion | 20 (7) | 2 (3) | 18 (8) | 0.327 | |
| Abrasion | 17 (6) | 4 (5) | 13 (6) | 0.954 | |
| Strain/sprain | 8 (3) | 2 (3) | 6 (3) | 1.038 | |
| Concussion | 2 (1) | 0 (0) | 2 (1) | | |
| Fracture | 1 (0) | 0 (0) | 1 (0) | | |
| Other injuries (e.g. soreness) | 6 (2) | 2 (3) | 4 (2) | | |
| Medical Treatment | 3 (1) | 0 (0) | 3 (1) | | |
| **Activity at time of fall:** | | | | | |
| Walking | 126 (42) | 27 (37) | 99 (44) | 0.739 | 0.394–1.464 |
| Sport | 111 (37) | 36 (49) | 75 (33) | 1.947 | 0.991–3.890 |
| Running/jogging | 30 (10) | 6 (8) | 24 (11) | 0.757 | 0.251–1.770 |
| Transfer | 10 (3) | 2 (3) | 8 (4) | 0.771 | |
| Other | 23 (8) | 2 (3) | 21 (9) | | |
| All activities except sport | 189 (63) | 37 (51) | 152 (67) | 0.507 | 0.264–1.027 |
| **Concurrent task:** | | | | | |
| No Concurrent Task | 194 (65) | 57 (78) | 137 (60) | 2.009 | 0.896–4.559 |
| Talking to friend | 74 (25) | 9 (12) | 65 (29) | **0.350** | **0.139–0.728** |
| Texting | 6 (2) | 1 (1) | 5 (2) | 0.616 | |
| Talking on phone | 2 (1) | 2 (3) | 0 (0) | | |
| Other | 24 (8) | 4 (5) | 20 (9) | | |
| **Environment:** | | | | | |
| Ice/slippery | 102 (34) | 25 (34) | 77 (34) | 1.015 | 0.568–1.969 |
| Nothing in environment | 64 (21) | 16 (22) | 48 (21) | 1.047 | 0.454–2.205 |
| Stairs (up and down) | 40 (13) | 7 (10) | 33 (15) | 0.582 | 0.166–1.461 |
| Footwear | 16 (5) | 1 (1) | 15 (7) | 0.196 | |
| Other | 90 (30) | 29 (40) | 61 (27) | | |
| **Alcohol/Drugs** | 33 (11) | 8 (11) | 25 (11) | 0.994 | 0.345–2.364 |
| **Lighting** | | | | | |
| Well lit | 197 (66) | 47 (64) | 150 (66) | 0.928 | 0.477–1.920 |
| Poorly lit | 97 (32) | 25 (34) | 72 (32) | 1.121 | 0.531–2.234 |
| **Outdoor vs. Indoor** | | | | | |
| Outdoor | 172 (57) | 43 (59) | 129 (57) | 1.089 | 0.539–2.231 |
| Indoor | 119 (40) | 28 (38) | 91 (40) | 0.930 | 0.423–1.875 |
| **Time of day** | | | | | |
| 6:00 AM–12:00 PM | 57 (19) | 8 (11) | 49 (22) | 0.608 | 0.238–1.294 |
| 12:00 PM–6:00 PM | 86 (29) | 23 (32) | 63 (28) | 1.197 | 0.594–2.401 |
| 6:00 PM–12:00 AM | 123 (41) | 32 (44) | 91 (40) | 1.166 | 0.633–2.118 |

(*Continued*)

**Table 2.** (Continued)

| | Number of Falls (percent) | | | OR | |
|---|---|---|---|---|---|
| | Total Fallers (N = 157, 300 falls) | Male Fallers (N = 43, 73 falls) | Female Fallers (N = 114, 227 falls) | | 95% CI |
| 12:00 AM–6:00 AM | 28 (9) | 8 (11) | 20 (9) | 1.274 | 0.430–3.301 |

Odds ratio (OR) and 95% confidence interval (CI) for males versus females (bivariate analysis). OR not calculated when either observation was zero, or when the category was other. CI not calculated when either observation was five or less, or when the category was 'other'. Significant difference bolded ($p<0.05$).

## Fall circumstances: Effect of sex

As noted in the methods, the effect of sex on fall circumstances was examined statistically only where there were sufficient observations to conduct the analyses. One circumstance was significantly different across sex: Females were significantly more likely to report talking to a friend (OR = 0.350, Table 2). While the remaining circumstances were not significantly different (or did not have sufficient observations for comparison), there were several trends that are noteworthy and should be considered for follow-up research with larger numbers of participants. Males demonstrated a trend where they were more likely to fall during sports activities than females (49% vs 33%; Table 2), and, similarly, females demonstrated a trend where they were more likely to fall during activities outside of sports (67% vs 51%; Table 2). Females demonstrated a trend of being more likely to fall on stairs (15% vs 10%; Table 2). Females had a trend of being more likely to report the fall as related to footwear (7% vs 1%; Table 2).

Regarding injuries, a trend was observed where females were more likely to report an associated injury than males (16% vs 10% of falls; Table 2). A fall-related injury was reported in 7% of all male participants and 12% of all female participants. Six males reported 10 injuries in 7 falls (10% of falls); 28 females reported 44 injuries in 36 falls (16% of falls) (Table 2). Males reported two contusions (29% of injuries), four abrasions (57%), and two sprain/strain injuries (29%). Females reported 18 contusions (50%), 13 abrasions (36%), six sprain/strain (16%), two concussions (6%), and one fracture (3%). In males, fall-related injuries occurred during sports (86% of injuries) and running (14% of injuries). In females, fall-related injuries occurred during walking (39% of injuries), sports (36%), running (19%), and standing/transfer (6%). Medical treatment was obtained for three falls (7% of all injuries, 1% of all falls); only females reported medical treatment.

## Slips and trips that did not result in a fall

During the 16 weeks, 2474 slips and 2483 trips that did not result in a fall were reported (average: 1.0 reported perturbations per week per participant, with 0.5 slips and 0.5 trips per week per participant). Two-hundred and thirty-seven participants (57 males, 180 females; 73% of all participants) reported at least one slip and 223 participants (50 males, 173 females; 60% of all

**Table 3. Frequency of reported slips and trips–that did not result in a fall–for males versus females.**

| | Number of Perturbations Without Falls (Percent of all Reports) | | | OR | 95% CI |
|---|---|---|---|---|---|
| | All | Male | Female | | |
| | (N = 325; 34,113 observations) | (N = 89; 9,395 observations) | (N = 236; 24,718 observations) | | |
| Slips | 2474 (7) | 719 (8) | 1755 (7) | 0.885 | 0.638–1.227 |
| Trips | 2483 (7) | 370 (4) | 2113 (9) | **0.446** | **0.297–0.670** |

Odds ratio (OR) and 95% confidence interval (CI) for males versus females. Significant differences bolded.

participants) reported at least one trip, where neither perturbation resulted in a fall. The reported slips were not different for males versus females ($p = 0.46$, Table 3), but females were more likely to report a trip ($p < 0.01$, Table 3) (average: 0.3 and 0.6 reported trips per week per male and female participant, respectively). Of all reported slips (with and without falls), 6% resulted in a fall, and of all reported trips, 4% resulted in a fall.

## Discussion

The primary purpose of this study was to determine if falls are associated with sex, physical activity level, and number of medications in young adults. Current knowledge was extended with a protocol that: (1) focused on a discrete age group, 18–27 years, and (2) included daily contemporaneous reports (i.e. prospective fall reports). Forty-eight percent of participants reported a fall, 14% of falls resulted in an injury, and 1% of falls required medical treatment. We observed that the frequency of falls was positively associated with physical activity level, number of medications, and being male. Females were more likely to be talking to a friend when a fall occurred and females were more likely to report a trip without a fall. Although there were not enough observations to run statistical analyses, only females sustained serious injuries (fractures and concussions) that required medical treatment.

Consistent with our previous findings [4] and recent research on fall-related injuries [1–3, 5, 6], falls in young adults were not uncommon and resulted in injuries. While about a third of these falls occurred due to participation in sports activities (where fall risk is higher), young adults also fell outside of sports as the following two observations emphasize. First, when sports-related falls were excluded, percent fallers remained relatively high at 39%. Second, while it seems reasonable to expect that the frequent fallers (25% of participants) were falling during sports, 83% of the frequent fallers fell at least once outside of sports (Fig 4B). Fall-related injuries were not uncommon in young adults: 10% of all participants reported an injury, 14% of falls resulted in an injury, while 1% of all participants and 1% of all falls resulted in injuries that were medically treated (one fracture and two concussions) (Table 2). Compared to a previous study in young adults (20–45 years, $n = 292$) [9], the percent fallers was more than twice as high (48 vs 18% of participants), and the percentage of falls with injuries was about five times lower (14 vs 72% of falls). The differences are likely due to the approach (prospective daily online survey vs retrospective report from past two years; retrospective reports are less accurate [17–20]), and the narrower range of ages observed here (18–27 vs 20–45 years). Overall, the results support the growing contention that falls are a serious concern for young adults [1–6].

The most common precipitating causes of falls were slips and trips (Table 2), which were also the most common causes of fall-related injuries in young adults [2] and are common causes of falls and fall-related injuries in older adults [19, 21–23]. Slips and trips were common in the young adults observed here, with an average of 0.5 slips and 0.5 trips per week in young adults (Table 3). However, young adults had sufficient balance skills to recover from the vast majority of these perturbations (94% of slips and 96% of trips did not result in a fall). These values are similar to slips in restaurant workers, where 0.4 slips were observed in 40 work hours, and workers recovered from 94% of slips [28]. The observation that trips occurred commonly in the field is consistent with observations in lab-based locomotor experiments: When stepping over stationary, visible obstacles in the lab, participants inadvertently contact the obstacle about 1–2% of the times they step over the obstacle (see summary in [29]). Overall, the observations reported here support the growing body of research that slips and trips are commonly experienced perturbations [4, 28, 29], and they are common causes of falls and injuries across the adult lifespan [2, 4, 19, 21–23]. These findings support the continued

development of perturbation training where people are trained to recover from falls by safely exposing them to slips and trips (e.g. [30]). Furthermore, the training may have benefit not only for balance-compromised populations, but also young healthy adults, since they regularly experience slips and trips and these perturbations are the most common cause of falls.

Higher levels of physical activity in young adults were associated with higher frequency of falling (Table 1), likely indicating increased exposure to hazardous situations [12]. Conversely, in older adults, higher levels of physical activity are generally associated with decreased falls, reflecting the positive effect of physical activity on age-related changes in balance (e.g. [13, 31, 32]). However, even in older adults, engaging in sports or vigorous-intensity physical activity was associated with increased falls [12, 31, 33]. Falls related to vigorous-intensity activity may be perceived as less important since these falls are hard to avoid and can be considered self-imposed. However, vigorous physical activities maximize health benefits (e.g. [34]), and national guidelines recommend moderate-to-vigorous activity in all ages [13]. Since vigorous activities are recommended for population health, it is important to understand the association between sports/vigorous-intensity activity and falls/fall-related injuries observed here and in earlier research [2, 9, 12, 31, 33]. Follow-up studies are recommended to (1) identify which sports and activities are more likely to lead to injuries, (2) determine possible steps for remediation within a sport or activity, and (3) identify individual factors that increase likelihood of injury during sports and vigorous activities.

A positive association was observed between number of falls and number of medications in this young adult sample (Fig 5), which is consistent with associations commonly observed in older adults (e.g. [15, 35]). This association has also been observed in a sample of young and mid-life adults (25–60 years) [16]; we demonstrate here that the association was also evident in a younger, narrower age range (18–27 years). While our study protocol does not allow us to determine if the association is due to underlying comorbidities or to the medications, it is evident that using one or more medications resulted in higher risk of falling. This is a largely unrecognized relationship in young adults that should be explored further and considered in fall prevention programs.

One of the goals of this study was to understand why young adult females sustain more fall-related injuries (Fig 1) [2, 3, 9]. While the number of injuries was not significantly higher in the current data, the trend indicated more injuries were observed in females (16% of females and 10% of males reported a fall-related injury, respectively; Table 2), consistent with previous research [2, 3, 9]. Higher injuries in females could result from the following possibilities: (1) females are more likely to be injured from a fall (where males and females fall at similar rates), and/or (2) females and males are equally likely to be injured from a fall, but females fall more frequently. The current observations do not support the idea that females sustain more injuries because they fall more frequently. Rather, we observed similar percentages of male and female participants that reported at least one fall (48% and 48%, respectively) and males reported a higher number of falls than females (Table 1). While males reported higher number of falls, we argue it is unlikely that females have better balance-ability relative to males based on several observations. These observations include: females reported twice as many trips without an associated fall, and females were more likely to be talking to a friend at the time of a fall than males. Furthermore, the following trends also do no support the idea that females have better balance than males: females were more likely to fall during lower risk activities, and females reported more injuries during walking (males reported no walking injuries). These observations are described more fully below.

The higher number of trips reported by females (Table 3) may indicate impaired gait control relative to males. Tripping results from inappropriate foot placement around an obstacle and/or insufficient foot elevation to clear the ground or an obstacle [29, 36, 37]. However, it is

important to note that since the number of trips were obtained from self-report, an alternative explanation is that females were more likely to *perceive* trips than males, rather than actually experiencing more trips. Perception may be affected by sex through factors such as higher levels of overconfidence in males [38]. We believe that the perception explanation is unlikely, because both sexes reported similar numbers of slips (Table 3), and it is unlikely that males would be more likely to perceive slips than trips. Therefore, it seems more likely that females tripped more, rather than females perceived more trips. Furthermore, a parallel is noted between the higher trip rate found here in young females and the observation that, across the adult lifespan, females were more likely than males to sustain an injury from a trip [2]. These parallel observations lead to the speculation that increased likelihood of tripping is a cross-age phenomenon for adult females (although the younger counterparts are less likely to sustain injury (Fig 1)).

Females were twice as likely as males to report talking at the time of the fall (Table 2), indicating that females may be more impaired during locomotor multitasking. Walking while talking is cognitively demanding not only for older adults [39], but also for young adults [40], due to the concurrent management of ongoing locomotion, terrain navigation, language formulation, and speech generation. This cognitive demand may be greater in females due to a differential focus on social versus physical cues [41, 42]. Road-crossing observations indicate that males focused more on physical cues (such as vehicles), while females focused more on social cues (such as other pedestrians), based on self-report measures [41] and gaze behavior [42]. If road crossing observations can be extrapolated to everyday tasks, males may attend more to the physical environment and be more likely to note upcoming hazards such as curbs and stairs, while females may attend more to social cues, such as the person they are talking to, and not sufficiently attend to the physical environment. Note that sex-related differences in gaze behavior may also explain why females are more likely to trip (Table 3), as females may not pay sufficient attention to the location and size of an obstacle. However, we note that higher reports of talking at the time of the fall by females has at least three possible explanations: 1) females were more balance-compromised than males when they were talking, as described above, 2) females were more likely to be talking than males during daily activities, and they happened to be talking when they fell, and/or 3) females were more likely to remember talking than males. Our protocol does not allow us to determine which possibility is accurate; lab-based research examining sex-related differences during locomotor multitasks is required.

The trends of (1) females falling during lower risk activities and (2) females reporting more injuries during walking (Table 2) are also consistent with impaired balance in females. The most common activity at the time of the fall for males was sports (49%), where falls are more likely to occur and perhaps even expected. Conversely, the most common activity for females was walking (44%), a lower-risk activity where falls are not expected in healthy adults. Although this trend was not statistically significant, it is consistent with previous research: The most common activity at the time of the fall was also sports-related for males and walking/ambulation for females (with fall self-report for ages 20–45 years [9] and with fall-injury medical reports for ages 18–44 years [2]). Therefore, consistent sex-related observations regarding activity at time of the fall for young adults were obtained in three different studies with different protocols, emphasizing that females are more likely to fall during lower risk activities, while males are more likely to fall during higher risk activities. Regarding activity during fall-related injuries, males reported no injuries during walking, while females reported that 38% of injuries occurred during walking (Table 2). All of the injuries in males occurred during the higher-risk activities of sports (86%) or while running/jogging (14%). Thus, there are a series of observations that support the idea that females have impaired balance/gait control relative

to males, although the results do not definitely identify why females sustain more injuries than males (Fig 1).

The sex-related differences leading to impaired balance/gait control in young adult females could result from the following: lower muscle mass and strength [43], slower and more variable reaction time [44], lower bone mass and skeletal integrity [45], greater focus on social cues (versus physical cues) [41, 42], and higher likelihood of wearing 'poor shoe types' (heels, sandals, and slippers) [46]. We also observed a trend where females were more likely to report footwear as causing the fall than males (Table 2). Overall, the differences for young adult females observed here lead to the recommendation that gait protocols should be developed to identify sex-related differences not only in older adults, but also in younger adults. Sex-related differences have been examined during steady state gait [47–51], but should also be extended to adaptive gait (e.g. obstacle crossing, stair climbing), gaze behavior during adaptive gait, perturbed gait, and perturbation training to further understand sex-related locomotor differences. In addition, the sex-related differences observed here indicate that fall prevention interventions will likely need to sex-specific. For example, consider the following observations from the current study and previous observations: 1) females reported a higher number of trips (Table 3), females were more likely than males to sustain an injury from a trip [2], and trips are a common cause of falls (Table 2; [19, 21–23]). These observations support further exploration of trips in females.

The high percentage of falls on stairs—13% of all falls, or 22% of non-sports related falls— are especially compelling given that traversing 60 flights of stairs a day accounts for only 1% of an 18-hour day (conservative estimate of 10 s per flight of stairs). Stairs are common locations for falls and fall-related injuries across the lifespan [2, 3, 52]. Half of the texting-related falls occurred on stairs, likely reflecting impaired gait behavior when attention is divided and/or gaze is diverted (e.g. [53–55]). The high percentage of falls on stairs observed here is consistent with the peak in injury rate due to stair-related falls for ages 21–30 years [3]. Furthermore, the trend for higher percentage of falls on stairs for females (15% vs 10%, Table 2) is consistent with higher injuries resulting from stair-related falls in females [3]. These observations demonstrate the need for continued research on stair behavior (e.g. [54, 56, 57]) as well as improved stair safety through environmental modifications (e.g. [58, 59]).

Consistent with research demonstrating that alcohol increases risks for falls (e.g. [60–63]), 11% of falls in the young adults observed here were reported as associated with alcohol/drugs (Table 2). The number of falls associated with alcohol/drugs may be higher than reported, due to the effects of intoxication on recall accuracy (e.g. [64]). Conversely, since college students are more likely to participate in heavy/binge drinking than their non-college attending peers [65–67], the alcohol-related falls reported here may be higher than that observed in the general population of young adults. Despite the clear association between alcohol and falls across the lifespan [5, 60–63], fall questionnaires/diaries do not always quantify the role of alcohol/recreational drugs (e.g. [2, 9, 19, 21–23]). Overall, fall-related research should include alcohol/recreational drugs when quantifying the circumstances of falls across the adult lifespan.

There are several limitations to this study. Self-report fall surveys are affected by recall bias [17, 18, 68, 69], but self-report is currently the only viable approach for examining fall frequency and circumstances in community-dwelling adults. For number of falls, we expect that daily surveys would mitigate the effect of recall bias on the number of falls [69]. However, reporting fall circumstances, such as what caused the fall, may have been affected by recall bias, as has been observed for older adults [68]. A promising future approach includes wearable devices that detect falls and daily activities, as the accuracy of these devices continues to improve (e.g. [70]). In low frequency measures, such as injuries, power may not have been adequate to detect sex-related differences. Similarly, the number of subjects who were taking three

or more medications was relatively low (18 out of 325 participants, Fig 5), but we note that the significant increase in number of falls was also observed from 0 to 1 medication, and from 1 and 2 medications. Follow-up research with larger numbers of participants is needed to ensure the observed associations are not spurious. Medication use was examined as number of prescription medications [15, 16], but more information may have been gained by examining categories of medications and dosages. Information on physical activity and medications was assessed once in the initial survey at baseline, but these measures may have changed over the observation period. For example, physical activity is affected by the change in seasons (e.g. [71, 72]). Finally, undergraduate students provide a convenience sample, but are not representative of all young adults aged 18–27 years. Future research should examine more heterogeneous groups.

## Summary

This study observed that frequency of falls was positively associated with physical activity level, number of prescription medications, and being male in young adults. While falls in older adults have been extensively examined, falls in young adults also occur frequently and should not be considered trivial. Since physical activity increased fall-risk in young adults, but is generally protective of falls in older adults, understanding the relationship between falls and physical activity across the lifespan is a critical area for developing interventions to mitigate falls in all adults. It is important to further explore the relationship between falls and number of medications given that it is consistent across the adult lifespan. The results here did not definitively identify why females sustain more fall-related injuries than males (Fig 1), but possibilities for further exploration were identified, including higher likelihood of tripping and impaired ability to walk while talking.

## Author Contributions

**Conceptualization:** HyeYoung Cho, Michel J. H. Heijnen, Shirley Rietdyk.

**Data curation:** HyeYoung Cho, Michel J. H. Heijnen.

**Formal analysis:** HyeYoung Cho, Michel J. H. Heijnen, Bruce A. Craig, Shirley Rietdyk.

**Supervision:** Shirley Rietdyk.

**Writing – original draft:** HyeYoung Cho, Shirley Rietdyk.

**Writing – review & editing:** HyeYoung Cho, Michel J. H. Heijnen, Bruce A. Craig, Shirley Rietdyk.

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
