## [Decision Letter · Decision Letter 0]

25 Nov 2020

PONE-D-20-33291

The frequency and circumstances of falls in young adults: the effect of sex, physical activity, and prescription medications

PLOS ONE

Dear Dr. Rietdyk,

Thank you for submitting your manuscript to PLOS ONE. After careful consideration, we feel that it has merit but does not fully meet PLOS ONE’s publication criteria as it currently stands. Therefore, we invite you to submit a revised version of the manuscript that addresses the points raised during the review process.

Both reviewers suggested major revisions to your manuscript. I agree with them. I believe that all suggestions can be addressed by the authors and would improve the manuscript. I ask attention for the suggestions/questions in the method section, especially about statistical analysis.

We look forward to receiving your revised manuscript.

Kind regards,

Fabio A. Barbieri, PhD

Academic Editor

PLOS ONE

Journal Requirements:

'Publication of this article was funded in part by Purdue University Libraries Open Access Publishing Fund.

The funders had no role in study design, data collection and analysis, decision to publish, or preparation of the manuscript.'

a. Please provide an amended statement that declares *all* the funding or sources of support (whether external or internal to your organization) received during this study, as detailed online in our guide for authors at http://journals.plos.org/plosone/s/submit-now

Please also include the statement “There was no additional external funding received for this study.” in your updated Funding Statement.

4. We noted in your submission details that a portion of your manuscript may have been presented or published elsewhere:

'The data from the first 94 young adult subjects were previously published in Human Movement Science (Heijnen & Rietdyk, 2016), but there were insufficient numbers of participants to identify the effects of prescription medication and sex. The submitted manuscript now includes 325 participants and demonstrates that, in young adults, the frequency and circumstances of falls are associated with sex, and the frequency of falls is associated with physical activity level and prescription medications. '

Please clarify whether this publication was peer-reviewed and formally published.

If this work was previously peer-reviewed and published, in the cover letter please provide the reason that this work does not constitute dual publication and should be included in the current manuscript.

Reviewers' comments:

Reviewer's Responses to Questions

**Comments to the Author**

1. Is the manuscript technically sound, and do the data support the conclusions?

Reviewer #1: Partly

Reviewer #2: Yes

2. Has the statistical analysis been performed appropriately and rigorously? 

Reviewer #1: Yes

Reviewer #2: Yes

3. Have the authors made all data underlying the findings in their manuscript fully available?

Reviewer #1: No

Reviewer #2: No

4. Is the manuscript presented in an intelligible fashion and written in standard English?

Reviewer #1: Yes

Reviewer #2: Yes

5. Review Comments to the Author

Reviewer #1: The purpose of this study was to determine if the frequency and circumstances of falls are associated with the sex, physical activity level, and number of medications in college-aged young adults.

Introduction:

Line 51, “The preceding observations were…” Please correct the citation. Citation 2 is not from medical reports but from a national survey using interviews.

Enough details on background and significance statements on why we are studying circumstances of medication and physical activity/sports related falls is not provided.

Methods:

Can you add how many males and females were approached for the survey? It looks like there is a systematic difference in the responses received by gender. This might have skewed the estimate due to selection bias more towards female as seen in the result. How did the investigators checked or accounted for this in the analyses?

Line 97: “The daily response rate ranged from 38-100%”. I could not compare this rate with the daily rates provided in Figure 2. However, the daily response rate is consistently higher than 80% (Figure 2). How was average response rate and daily response rate computed? When calculating daily response rate, did you use it among those who responded as the denominator?

Initial survey:

The study took over a period of 16 weeks. How likely is it that the number of prescription medication and the weekly physical activity would change? Change in these variables might influence the falls reported over time. Since these information were collected at baseline (week 1), it seems that the analysis did not take into account for this and hence to the least this has to be acknowledged in the limitation.

Statistical Analyses:

Please provide rationale for the use of mean centered LTEQ and number of medications and also the response days as the offset variable.

When defining the use of GLMM, please provide the covariance structure and its justification used to account for the correlation in falls among repeated fallers. Though it is understood that logit link function was used in GLMM, please specify the link function in the methods.

Line 166: The investigators choose to not calculate OR for samples less than 5 or for categories “other”. Did the author do sensitivity check by using some exact methods or adjustments using “sandwich variance estimation” or its variants to account for small sample sizes?

It is not clearly mentioned if multivariable analysis was done and what/how were the variables selected for the multivariable model (both Poisson and zero-inflated components)? Multivariable analysis will be able to account for several confounding effect and is strongly recommended to add it to the manuscript.

Results:

Line 199-200: Please specify the details of the interaction model and AIC used for ZIP in the method section.

Line 200-203: Please specify the parameter estimates and 95% CI for Poisson and zero-inflated components.

Table 1: Are these ORs from bivariate analyses? Please specify as footnote and also explicitly in the methods.

Discussions:

The discussion section includes lot of results rather than discussing and telling the story why the investigators observed the way that their study did and how does it compares and contrasts with the previous studies. What could be the potential causes that would be an alternative explanation for the contrasts?

Limitations: What would be the anticipated quantification of the recall bias that comes into play when the survey is completed daily for the falls in the past 24 hours? I would not call it a recall bias unless the investigators think there is a substantial recall bias.

Reviewer #2: This study aimed to quantify the frequency and circumstances of falls in young adults. The research is relevant, original and offers a contribution to the knowledge of the area. In that sense, I congratulate the authors. The manuscript, however, needs to be revised and some suggestions and questions are presented here for good application of findings.

1. The authors did a great job in collecting data from a significant number of participants (N=325). Why did the authors include some restrictions in allowing full access to the data?

Title:

2. Adequate

Abstract:

3. The abstract is well done. It has 295 words, close to the limit of 300 as stipulated by Plos One. I wonder if it is possible to include the statistical tests used in this study. In addition, I’m not sure if the sentence “Higher number of falls were associated with the following: …. and being male (p=0.008)” is contemplated in the conclusion of the abstract.

Introduction:

4. Do authors need any prior authorization from WISQARS to publish figure 1?

5. The use of classical references is important. However, those references should not represent 50% of the references included in the introduction section. Please update the introduction section with recent papers (2016-2021).

6. Rows 63-64: Exclude “Table 2”.

7. Some parts of the introduction are confusing and must be rewritten.

8. Please include a last paragraph with authors’ hypotheses.

Methods:

9. Did the authors perform the sample size calculation? I understand that all students were invited to participate in this research, but a sample size calculation is important to ensure that the statistical errors (type 1 - alpha and type 2 - beta) were controlled.

10. I suggest the use of a flowchart to clarify readers about inclusions and exclusions.

11. Did the authors obtain information about participants’ diseases, amputations, lower limb prostheses, vision problems, vertigo?

12. Statistical analyses: appropriate and well described!

13. I am not sure if the students know the differences between concussion and contusion.

Results:

14. The results are well described!

Discussions:

15. Is it be possible to analyze which sports caused the most injuries/falls?

16. Are students who do not practice sports predisposed to having less falls?

17. Include references supporting the statements seen on rows 396 to 401.

18. Please include discussion about the impact of drugs and alcohol on falls.

19. Previous diseases were reported by students?

Summary:

20. Adequate

References:

21. Most of references were published more than 5 years ago. Authors should include more recent publications.

6. PLOS authors have the option to publish the peer review history of their article (what does this mean?). If published, this will include your full peer review and any attached files.

Reviewer #1: No

Reviewer #2: **Yes: **Gustavo Christofoletti

---

## [Author Response · Author response to Decision Letter 0]

26 Feb 2021

We thank the reviewers for their comments, we believe that edits made to address the comments have increased clarity. Regarding the statistics, we did conduct a multivariable analysis (effect of sex, physical activity and number of medications on number of falls) for frequency of falls, and this is now more clearly described in the methods and results. We also changed our OR analyses for the fall circumstances due to participants submitting more than one fall report, as outlined in responses 6&7 for reviewer 1. We also clarified the purpose of the study: our primary purpose was to quantify the association between falls and sex, physical activity, and number of medications. The other analyses (circumstances of falls and slips and trips that did not result in a fall) support the interpretation of these findings. 

We also apologize for the confusion regarding availability of the data, it is available here: http://bit.ly/3dQikcG

Reviewer #1: The purpose of this study was to determine if the frequency and circumstances of falls are associated with the sex, physical activity level, and number of medications in college-aged young adults.

Introduction:

COMMENT 1: Line 51, “The preceding observations were…” Please correct the citation. Citation 2 is not from medical reports but from a national survey using interviews.

Enough details on background and significance statements on why we are studying circumstances of medication and physical activity/sports related falls is not provided.

RESPONSE 1: Thank you for finding this mistake – we have edited the text to accurately describe the contribution of the cited studies. See lines 41-42. 

We have added details on the importance of studying the association between falls and medication, as well as the association between falls and physical activity levels. For falls and physical activity, see lines 64-71. For falls and medication, see lines 73-78. 

Methods:

COMMENT 2: Can you add how many males and females were approached for the survey? It looks like there is a systematic difference in the responses received by gender. This might have skewed the estimate due to selection bias more towards female as seen in the result. How did the investigators checked or accounted for this in the analyses?

RESPONSE 2: In order to quantify the males and females approached, we would need to review information of students who did not agree to participate in the study. Therefore, we checked with the university’s IRB, and they confirmed that we are not allowed to reveal any information regarding students who were approached, but did not participate. However, we can report that the undergraduates enrolled in our department (Health and Kinesiology) are 64% female, consistent with the higher percentage of female participants (72%) observed in the study. We point out that we are comparing the mean for the females against the mean for the males, thus, the size of each sample should not affect the estimated sex effect. 

COMMENT 3: Line 97: “The daily response rate ranged from 38-100%”. I could not compare this rate with the daily rates provided in Figure 2. However, the daily response rate is consistently higher than 80% (Figure 2). How was average response rate and daily response rate computed? When calculating daily response rate, did you use it among those who responded as the denominator?

RESPONSE 3: We apologize for the confusion – we did not calculate daily response rate. Response rate was calculated for each participant as number of responses/number of surveys, and then the response rates were averaged across participants. Response rate was calculated (1) for the 16 week observation period, and (2) as weekly response rate to demonstrate the change over time (Figure 2). To increase clarity, we added the response rate equation, as well as the mean, median, and mode of the response rate (91, 98, and 100%, respectively). We also indicated that 6 participants did not complete any daily surveys after signing the informed consent. Please see lines 102-104 and the caption for Figure 3. 

Initial survey:

COMMENT 4: The study took over a period of 16 weeks. How likely is it that the number of prescription medication and the weekly physical activity would change? Change in these variables might influence the falls reported over time. Since these information were collected at baseline (week 1), it seems that the analysis did not take into account for this and hence to the least this has to be acknowledged in the limitation.

RESPONSE 4: Thank you for bringing this issue to our attention. For physical activity, students were asked to report on a typical week. Although students have access to indoor recreational facilities, so they can be active regardless of weather, it is possible that amount of physical activity changed with the seasons. It is also possible that their medications changed over the 16 weeks. We have added these details to the limitations section, please see added text below (in italics) and lines 526-528. 

Limitations paragraph: Information on physical activity and medications was assessed once at baseline, but these measures may have changed over the observation period. For example, physical activity is affected by the change in seasons (e.g. Yoshimura et al., 2020; Westerterp, 2020). 

Statistical Analyses:

COMMENT 5: Please provide rationale for the use of mean centered LTEQ and number of medications and also the response days as the offset variable.

RESPONSE 5: The variables were centered in order to reduce the collinearity between the interaction term and the associated main effect terms. The number of response days was selected as an offset variable to account for the differences in sampling intensity. In other words, we are normalizing the data and modeling the rate (number of falls per day) rather than the actual count. 

The preceding text has been added to the methods as requested, see lines 171-174. 

COMMENT 6: When defining the use of GLMM, please provide the covariance structure and its justification used to account for the correlation in falls among repeated fallers. Though it is understood that logit link function was used in GLMM, please specify the link function in the methods.

RESPONSE 6: Based on this and other comments of yours, we reviewed our modeling approach for Table 2 and determined that we should take a conditional model approach focusing just on the 300 fall reports from 157 individuals. Thus, we moved away from the GLMM model and instead considered 2x2 contingency table analyses. To account for the fact that a subject could contribute data on more than one fall, the odds ratio confidence intervals were constructed using a bias-corrected and accelerated bootstrap interval. In addition to removing the description of the GLMM model, the following text has been added: “A bias-corrected and accelerated (BCa) bootstrap interval of the odds ratio (OR; males divided by females) was constructed (bivariate analysis). The BCa bootstrap interval of the OR was used to account for the fact that each participant could contribute multiple falls (Fig 2).” See lines 175-182. 

COMMENT 7: Line 166: The investigators choose to not calculate OR for samples less than 5 or for categories “other”. Did the author do sensitivity check by using some exact methods or adjustments using “sandwich variance estimation” or its variants to account for small sample sizes?

RESPONSE 7: As noted in the previous comment, we adjusted our OR calculation. With this new analysis, one circumstance was significantly affected by sex: Females were more likely to report talking to a friend at the time of the fall. While this approach, to some degree handles small sample sizes, we compared the interval to the interval using Wald modified and Exact methods. The outcome was roughly the same. Therefore, for parsimony, we only reported the bootstrap OR calculation. As to the comment regarding cell count less than 5, we found the bootstrap CI to be too wide to be of use and simply a distraction. 

COMMENT 8: It is not clearly mentioned if multivariable analysis was done and what/how were the variables selected for the multivariable model (both Poisson and zero-inflated components)? Multivariable analysis will be able to account for several confounding effect and is strongly recommended to add it to the manuscript.

RESPONSE 8: We apologize for the confusion, we did complete multivariable analysis. We have added the following text to the methods to clarify: “The model included two and three-factor interactions (sex, physical activity, and number of medications) and we used AIC to select the best model..” See methods, lines 168-170 and results, lines 216-217.

Results:

COMMENT 9: Line 199-200: Please specify the details of the interaction model and AIC used for ZIP in the method section.

RESPONSE 9: Text added as requested; this issue overlapped with the preceding comment (#8), and the same addition addressed both the preceding and current comment. See lines 168-170.

COMMENT 10: Line 200-203: Please specify the parameter estimates and 95% CI for Poisson and zero-inflated components.

RESPONSE 10: We added a new Table with the parameter estimates and 95% CI, see Table 1 (also included below). 

 Zero-Inflation Component Poisson Component

 Estimate (SE) 95% CI Lower 95% CI Upper p Estimate (SE) 95% CI Lower 95% CI Upper p

Intercept -0.3066 (0.1785) -0.6564 0.0432 0.0858 -3.9832 (0.0775) -4.1351 -3.8313 <0.0001

Medications 0.3214 (0.1429) 0.0413 0.6015 0.0245 0.3367 (0.0519) 0.2349 0.4384 <0.0001

Sex 0.3529 (0.1328) 0.0926 0.6132 0.0079

Physical Activity 0.0041 (0.0018) 0.0005 0.0077 0.0250

COMMENT 11: Table 1: Are these ORs from bivariate analyses? Please specify as footnote and also explicitly in the methods.

RESPONSE 11: Yes, the ORs are from bivariate analyses. This is now specified in the Table caption and in the methods, see line 263 (Table caption) and lines 179 (methods).

Discussions:

COMMENT 12: The discussion section includes lot of results rather than discussing and telling the story why the investigators observed the way that their study did and how does it compares and contrasts with the previous studies. What could be the potential causes that would be an alternative explanation for the contrasts?

RESPONSE 12: The discussion section has been edited to remove the results and focus on the interpretation and comparison with previous studies. All new text is highlighted; you will see that it includes most of the discussion. 

COMMENT 13: Limitations: What would be the anticipated quantification of the recall bias that comes into play when the survey is completed daily for the falls in the past 24 hours? I would not call it a recall bias unless the investigators think there is a substantial recall bias.

RESPONSE 13: Thank you for allowing us to clarify this point – we do not expect recall bias in whether or not a person fell, but rather in the circumstances of the fall. Please see edit below (in italics) and on lines 514-517.

For number of falls, we expect that daily surveys would mitigate the effect of recall bias on the number of falls (Hannan et al., 2010). However, reporting fall circumstances, such as what caused the fall, may be affected by recall bias, as has been observed for older adults (Yang et al., 2015). 

Yang Y, Feldman F, Leung PM, Scott V, Robinovitch SN. Agreement between video footage and fall incident reports on the circumstances of falls in long-term care. Journal of the American Medical Directors Association. 2015 May 1;16(5):388-94.

Reviewer #2: This study aimed to quantify the frequency and circumstances of falls in young adults. The research is relevant, original and offers a contribution to the knowledge of the area. In that sense, I congratulate the authors. The manuscript, however, needs to be revised and some suggestions and questions are presented here for good application of findings.

RESPONSE: Thank you for the support of our research. 

COMMENT 1. The authors did a great job in collecting data from a significant number of participants (N=325). Why did the authors include some restrictions in allowing full access to the data?

RESPONSE 1: We do not intend to restrict any of the data (other than identifiable information), it seems the wrong option was selected at the time of submission. The data is available at the following website: 

http://bit.ly/3dQikcG

Title:

COMMENT 2. Adequate

RESPONSE 2: Please note that we have modified the title. Previous title: The frequency and circumstances of falls in young adults: the effect of sex, physical activity, and prescription medications. New title: Falls in young adults: The effect of sex, physical activity, and prescription medications. 

Abstract:

COMMENT 3. The abstract is well done. It has 295 words, close to the limit of 300 as stipulated by Plos One. I wonder if it is possible to include the statistical tests used in this study. In addition, I’m not sure if the sentence “Higher number of falls were associated with the following: …. and being male (p=0.008)” is contemplated in the conclusion of the abstract.

RESPONSE 3: As requested, we added the statistical test to the abstract (zero-inflated Poisson model) (see line 27). We completed minor edits to reduce the word count so that we could extend the conclusion as requested, with the following text in italics and on lines 34-37: 

“In conclusion, the rate of falls in young adults was affected by physical activity levels, number of medications, and sex. Quantifying and understanding these differences leads to increased knowledge of falls across the lifespan and is instrumental in developing interventions to prevent falls.” 

COMMENT 4. Do authors need any prior authorization from WISQARS to publish figure 1?

RESPONSE 4: The data on WISQARS are publicly available, but researchers are required to provide a citation, as we have done in the figure caption: https://www.cdc.gov/injury/wisqars/fatal_help/faq.html#types

COMMENT 5. The use of classical references is important. However, those references should not represent 50% of the references included in the introduction section. Please update the introduction section with recent papers (2016-2021).

RESPONSE 5: As requested, we have added the following 12 papers that are more recent. However, we did not remove any of the older references, as we respectfully argue that the cited literature should be chosen based on the relevance to the topic, rather than the year of publication. For example, while there is plenty of recent research regarding medication and falls in older adults, we could only find one article on medications and fall-risk in young adults, and it was published in 2012 (Kool et al., 2012). We had similar issues with other topics. To follow up further, we randomly picked a recent paper from the citation list: Blazewick et al., 2018. This paper examined fall-related injuries on stairs, and only 10 of 47 references were dated 2013 or later (for 2018 publication date). Thus, we believe that the additions we made are sufficient. 

1. Allin LJ, Brolinson PG, Beach BM, Kim S, Nussbaum MA, Roberto KA, et al. Perturbation-based balance training targeting both slip- And trip-induced falls among older adults: A randomized controlled trial. BMC Geriatr. 2020;20: 1–13. 

2. Chen CM, Yoon YH. Usual Alcohol Consumption and Risks for Nonfatal Fall Injuries in the United States: Results From the 2004–2013 National Health Interview Survey. Subst Use Misuse. 2017;52: 1120–1132. 

3. Cho H, Romine NL, Barbieri FA, Rietdyk S. Gaze diversion affects cognitive and motor performance in young adults when stepping over obstacles. Gait Posture. 2019;73: 273–278. 

4. Jacobs J V. A review of stairway falls and stair negotiation: Lessons learned and future needs to reduce injury. Gait Posture. 2016;49: 159–167. 

5. Lee J. The association between physical activity and risk of falling in older adults: A systematic review and meta-analysis of prospective cohort studies. Geriatr Nurs. 2020;41. 

6. McErlean DR, Hughes JA. Who falls in an adult emergency department and why—A retrospective review. Australas Emerg Nurs J. 2017;20: 12–16. 

7. Muir BC, Bodratti LA, Morris CE, Haddad JM, van Emmerik REA, Rietdyk S. Gait characteristics during inadvertent obstacle contacts in young, middle-aged and older adults. Gait Posture. 2020;77: 100–104. 

8. Piercy KL, Troiano RP, Ballard RM, Carlson SA, Fulton JE, Galuska DA, et al. The physical activity guidelines for Americans. JAMA - J Am Med Assoc. 2018;320: 2020–2028. 

9. Schreiber Compo N, Carol RN, Evans JR, Pimentel P, Holness H, Nichols-Lopez K, et al. Witness memory and alcohol: The effects of state-dependent recall. Law Hum Behav. 2017;41: 202–215. 

10. Thomas NM, Skervin T, Foster RJ, O’Brien TD, Carpenter MG, Maganaris CN, et al. Optimal lighting levels for stair safety: Influence of lightbulb type and brightness on confidence, dynamic balance and stepping characteristics. Exp Gerontol. 2020;132: 110839. 

11. Westerterp KR. Seasonal variation in body mass, body composition and activity-induced energy expenditure: a long-term study. Eur J Clin Nutr. 2020;74: 135–140. 

12. Yoshimura E, Tajiri E, Hatamoto Y, Tanaka S. Changes in Season Affect Body Weight, Physical Activity, Food Intake, and Sleep in Female College Students: A Preliminary Study. Int J Environ Res Public Health. 2020;17: 8713. 

COMMENT 6. Rows 63-64: Exclude “Table 2”.

RESPONSE 6: Text edited as requested. 

COMMENT 7. Some parts of the introduction are confusing and must be rewritten.

RESPONSE 7: We reviewed the introduction carefully, and edited the text to increase clarity, and also to address a comment by Reviewer 1 (comment 1). Please see the highlighted text throughout the introduction. 

COMMENT 8. Please include a last paragraph with authors’ hypotheses.

RESPONSE 8: Since this was an exploratory study, rather than an experimental study, we did not include specific hypotheses. This approach (i.e. not including hypotheses) is consistent with other manuscripts that have quantified falls and/or fall-related injuries (Blazewick et al., 2018; Stevens & Sogolow, 2005; Talbot, Musiol, Witham, & Metter, 2005; Timsina et al., 2017; Verma et al., 2016). 

Methods:

COMMENT 9. Did the authors perform the sample size calculation? I understand that all students were invited to participate in this research, but a sample size calculation is important to ensure that the statistical errors (type 1 - alpha and type 2 - beta) were controlled.

RESPONSE 9: We did not perform a sample size calculation. This was an exploratory study rather than a designed study with specific hypotheses. To our knowledge, this is the first time that falls have been examined as a function of physical activity, medications, and sex in young adults (while Talbot et al. (2005) examined the effect of sex on falls in young adults, they did not consider physical activity or medication). Thus, we did not have data to conduct sample size calculations. Our approach was to collect as much information as possible in the time window we had. The resultant sample size may be too small to find some effects (Type II error) but it can now be used to help in the design of future studies. Replication is needed to determine if the observations are robust or not. 

COMMENT 10. I suggest the use of a flowchart to clarify readers about inclusions and exclusions.

RESPONSE 10: Thank you for this suggestion – we agree that the flowchart helps with clarity. Please see Figure 2 in the document. 

COMMENT 11. Did the authors obtain information about participants’ diseases, amputations, lower limb prostheses, vision problems, vertigo?

RESPONSE 11: In the first version of survey, we did not ask about diseases or disorders. We updated the survey to include this information in a later version. Therefore, 171 of the 325 participants (53%) have this information. Since we did not have this information on all participants, we did not report it. If you are curious, here are the details: Methods: Participants were provided a list of common diseases/disorders, and an option called ‘Other’ that they could fill in anything not included on the list. We observed the following in the 171 participants who received this question: Depression (n = 14, 5 fallers), insomnia (n = 6, 3 fallers), diabetes (n = 2, 1 faller), arthritis (n = 2, 0 fallers), COPD (n = 1, 1 faller), inner ear disorder (n = 1, 0 fallers). Under ‘other’, the following were given: scoliosis, hypoglycemia, celiac disease, asthma, epilepsy, concussion. For visual disorders, 1 person reported cataracts.

COMMENT 12. Statistical analyses: appropriate and well described!

RESPONSE 12: Thank you! Although note that additional information and analyses were added based on comments 5-11 of Reviewer 1.

COMMENT 13. I am not sure if the students know the differences between concussion and contusion.

RESPONSE 13: When students took the survey, contusion was defined as a bruise, to reduce any confusion. However, since these were university students, we are confident that they understood the difference between a concussion and a contusion. When the survey is disseminated to non-university students, it will be a good idea to add definitions as needed. 

Results:

COMMENT 14. The results are well described!

RESPONSE 14: Thank you!

Discussions:

COMMENT 15. Is it be possible to analyze which sports caused the most injuries/falls?

RESPONSE 15: We are interested in this question, but do not have sufficient data to answer fully. When students identified ‘sports’ as the activity at the time of the fall, we did ask them to identify the sport, and the highest number of sports-related falls occurred in soccer (21 falls), basketball (18 falls), and football (11 falls) (note that soccer in North America would be called football in the rest of the word). Other sports with falls included volleyball, rock climbing, skateboarding, racquetball, lacrosse, throwing hammer or discuss. However, we do not have a measure of sports participation. Thus, we do not know if they fell more during soccer because the sport has a higher fall-risk or because soccer is a commonly played sport. Since we do not have this information to allow us to normalize sport participation, we did not report the sports with the most falls. However, we note that falls as a function of sports should be addressed in future research, please see lines 372-386. 

COMMENT 16. Are students who do not practice sports predisposed to having less falls?

RESPONSE 16: We are very interested in this question! Unfortunately, in this study, we measured physical activity in general, but did not quantify participation in individual sports. Therefore, with the current survey, we cannot answer this question. We are currently exploring sport participation surveys to include in future research. 

COMMENT 17. Include references supporting the statements seen on rows 396 to 401.

RESPONSE 17: We apologize for the confusion. Please note that the statements on rows 369 to 401 (note that the statements have moved and are now on lines 446-449) are possible situations or behaviors that we believe may have occurred in order to explain the observed results. We observed that females were twice as likely to report talking as a concurrent task at the time of the fall. Thus, we developed a list of possibilities that could result in that outcome, and there are not any associated references. 

COMMENT 18. Please include discussion about the impact of drugs and alcohol on falls.

RESPONSE 18: Text has been added to the discussion as requested. See lines 501-510.

COMMENT 19. Previous diseases were reported by students?

RESPONSE 19: This appears to be a duplicate of Comment 11; please see answer in Response 11 above.

Summary:

COMMENT 20. Adequate

RESPONSE 20: No response needed.

References:

COMMENT 21. Most of references were published more than 5 years ago. Authors should include more recent publications.

RESPONSE 21: This appears to be a duplicate of Comment 5; please see answer in Response 5 above.

---

## [Decision Letter · Decision Letter 1]

6 Apr 2021

Falls in young adults: the effect of sex, physical activity, and prescription medications

PONE-D-20-33291R1

Dear Dr. Rietdyk,

We’re pleased to inform you that your manuscript has been judged scientifically suitable for publication and will be formally accepted for publication once it meets all outstanding technical requirements.

Kind regards,

Fabio A. Barbieri, PhD

Academic Editor

PLOS ONE

Additional Editor Comments (optional):

Reviewers' comments:

Reviewer's Responses to Questions

**Comments to the Author**

1. If the authors have adequately addressed your comments raised in a previous round of review and you feel that this manuscript is now acceptable for publication, you may indicate that here to bypass the “Comments to the Author” section, enter your conflict of interest statement in the “Confidential to Editor” section, and submit your "Accept" recommendation.

Reviewer #1: All comments have been addressed

Reviewer #2: All comments have been addressed

2. Is the manuscript technically sound, and do the data support the conclusions?

Reviewer #1: Yes

Reviewer #2: Yes

3. Has the statistical analysis been performed appropriately and rigorously? 

Reviewer #1: Yes

Reviewer #2: Yes

4. Have the authors made all data underlying the findings in their manuscript fully available?

Reviewer #1: Yes

Reviewer #2: Yes

5. Is the manuscript presented in an intelligible fashion and written in standard English?

Reviewer #1: Yes

Reviewer #2: Yes

6. Review Comments to the Author

Reviewer #1: There is one minor suggestion:

Please add a sentence along the line of your response: "the undergraduates enrolled in our department (Health and Kinesiology) are 64% female, consistent with the higher percentage of female participants (72%) observed in the study." in describing the population of your study to highlight the representation of the sample and to avoid confusion about higher representation of the female population in this study.

Reviewer #2: I congratulate the authors for reviewing the manuscript. All my questions were answered and, in my point of view, the article has merits and qualities for publication in its current version.

7. PLOS authors have the option to publish the peer review history of their article (what does this mean?). If published, this will include your full peer review and any attached files.

Reviewer #1: No

Reviewer #2: **Yes: **Gustavo Christofoletti

---

## [Editor Report · Acceptance letter]

13 Apr 2021

PONE-D-20-33291R1 

Falls in young adults: The effect of sex, physical activity, and prescription medications 

Dear Dr. Rietdyk:

I'm pleased to inform you that your manuscript has been deemed suitable for publication in PLOS ONE. Congratulations! Your manuscript is now with our production department. 

Kind regards, 

on behalf of

Dr. Fabio A. Barbieri 

Academic Editor

PLOS ONE